# Metabolic Effects of Resistant Starch Type 2: A Systematic Literature Review and Meta-Analysis of Randomized Controlled Trials

**DOI:** 10.3390/nu11081833

**Published:** 2019-08-08

**Authors:** Matthew Snelson, Jessica Jong, Deanna Manolas, Smonda Kok, Audrey Louise, Romi Stern, Nicole J. Kellow

**Affiliations:** 1Department of Diabetes, Central Clinical School, Monash University, Melbourne, VIC 3004, Australia; 2Department of Nutrition, Dietetics & Food, Monash University, Level 1, 264 Ferntree Gully Road, Notting Hill, VIC 3168, Australia

**Keywords:** resistant starch, dietary fiber, obesity, metabolic syndrome, type 2 diabetes, metabolic health, systematic review

## Abstract

Published evidence exploring the effects of dietary resistant starch (RS) on human cardiometabolic health is inconsistent. This review aimed to investigate the effect of dietary RS type 2 (RS2) supplementation on body weight, satiety ratings, fasting plasma glucose, glycated hemoglobin (HbA1c), insulin resistance and lipid levels in healthy individuals and those with overweight/obesity, the metabolic syndrome (MetS), prediabetes or type 2 diabetes mellitus (T2DM). Five electronic databases were searched for randomized controlled trials (RCTs) published in English between 1982 and 2018, with trials eligible for inclusion if they reported RCTs involving humans where at least one group consumed ≥ 8 g of RS2 per day and measured body weight, satiety, glucose and/or lipid metabolic outcomes. Twenty-two RCTs involving 670 participants were included. Meta-analyses indicated that RS2 supplementation significantly reduced serum triacylglycerol concentrations (mean difference (MD) = −0.10 mmol/L; 95% CI −0.19, −0.01, *P* = 0.03) in healthy individuals (*n* = 269) and reduced body weight (MD = −1.29 kg; 95% CI −2.40, −0.17, *P* = 0.02) in people with T2DM (*n* = 90). However, these outcomes were heavily influenced by positive results from a small number of individual studies which contradicted the conclusions of the majority of trials. RS2 had no effects on any other metabolic outcomes. All studies ranged from 1–12 weeks in duration and contained small sample sizes (10–60 participants), and most had an unclear risk of bias. Short-term RS2 supplementation in humans is of limited cardiometabolic benefit.

## 1. Introduction

Obesity, characterised by extreme excess adiposity, has been recognized as a global epidemic [1]. It is well established that overweight and obesity (body mass index (BMI) ≥ 25 kg/m^2^) have implications for the development of metabolic perturbations such as the metabolic syndrome (MetS) and type 2 diabetes mellitus (T2DM) [2,3]. Approximately 25% of the world’s adults have MetS, which incurs a 5-fold greater risk of developing type 2 diabetes [4]. Similarly, T2DM has been identified as a growing epidemic, affecting an estimated 422 million individuals worldwide in 2015, and its prevalence is expected to rise to 642 million by 2040 [5]. Environmental factors involved in the etiology of MetS and T2DM include the increasing global rates of obesity, sedentary lifestyles and the consumption of energy-dense, nutrient-depleted, highly refined foods. While there is much focus on the negative health effects of consuming excessive quantities of nutrients such as saturated fat, sugar and sodium, an additional consequence of increased consumption of highly processed foods is the displacement of beneficial nutrients from the diet. Dietary fiber is one food component which is consumed in insufficient quantities [6], despite the growing body of evidence to support its health benefits in the prevention and treatment of chronic diseases.

Dietary fiber can be broadly defined as those carbohydrate polymers and oligomers (plus lignin) which escape digestion in the small intestine, passing into the large bowel where they are partially (insoluble dietary fiber) or more completely (soluble dietary fiber) fermented and metabolised by the gut microbiota [7]. Recognized health benefits associated with dietary fiber consumption include improved gastrointestinal function, moderation of circulating blood lipids and attenuation of post-prandial serum glucose and insulin responses [8]. The non-starch polysaccharides (NSPs) which form important structural components of plant cell walls constitute a significant proportion of human dietary fiber intake. Inadequate consumption of dietary fiber is a major concern in Western societies, with an estimated 90% of adults failing to consume the recommended total of 25–35 g of fiber per day [9]. Chronic low intake of dietary fiber has been associated with many deleterious health consequences including increased risk for colorectal cancer, diverticular disease, cardiovascular disease, obesity, the metabolic syndrome, pre-diabetes, type 2 diabetes and poorly controlled diabetes [10,11,12,13,14,15]. An analysis of 17 prospective cohort studies reported that every 2 g increase in cereal fiber intake per day was associated with a 6% reduction in the risk of developing type 2 diabetes [16].

The prebiotic properties of long-chain non-viscous fermentable fibers such as inulin-type fructans, galactans and resistant starch (RS) have been the focus of recent research. The bacterial fermentation products of these fibers are thought to provide a variety of localized and systemic health benefits to the host, some of which may assist in the prevention and management of obesity, MetS and T2DM. Resistant starches are potential prebiotics defined as any starch which resists digestion in the small intestine and is fermented by bacteria upon reaching the large intestine [17]. Resistant starches have been classified into five categories. Resistant starch type 1 (RS1) is found in whole grains and legumes that are inaccessible to digestive enzymes as they are surrounded by a protective barrier. Resistant starch type 2 (RS2), the most widely studied form of RS, contains dense ungelatinized starch granules which inhibit enzyme access and activity, and include high amylose corn starch and raw potatoes. Retrograded starches, such as cooked and cooled potatoes, rice and pasta are categorized as resistant starch type 3 (RS3). Retrogradation occurs when starches are first heated to undergo gelatinization, and then cooled to form a crystalline structure. Resistant starch type 4 (RS4) is formed via chemical cross-linking of starch by the addition of esters and ether groups. Lastly, resistant starch type 5 (RS5) forms when amylose and long branch chains of amylopectin form single-helical complexes with fatty acids and fatty alcohols, preventing enzyme access to the starch [18]. Consumption of different types of RS appears to result in unique physiological outcomes in human studies [19], and therefore the potential health effects of each RS category must be evaluated individually.

The exact mechanisms underlying proposed cardiometabolic benefits of RS remain elusive. However, it has been suggested that they may be mediated by compounds such as short-chain fatty acids (SCFAs) that are produced during the microbial fermentation of RS [20,21,22]. RS is selectively fermented in the large intestine by butyrate-producing bacteria, including *Eubacterium rectale* and *Bifidobacterium* species [23]. The SCFA butyrate is important for the maintenance of the health and integrity of the gastrointestinal tract and may play a role in modulating glucose and lipid homeostasis [24]. While animal studies have demonstrated positive effects of dietary RS supplements on body weight [25], insulin sensitivity [26], lipid levels [27] and inflammation [28], the doses of RS administered to rodents have far exceeded that which could realistically be consumed by humans.

To the best of our knowledge, no systematic reviews have explored the impact of RS2 consumption on metabolic outcomes in healthy or overweight individuals, or those with MetS or T2DM. This review aims to summarize the outcomes of published randomized controlled trials (RCTs) investigating the effects of dietary RS2 supplementation (≥8 g/day) in comparison to a placebo on body weight, appetite and markers of glucose and lipid metabolism in healthy individuals, those who are overweight/obese or have MetS or T2DM.

### 1.1. Methods

#### 1.1.1. Study Identification

This review was conducted in accordance with the Preferred Reporting Items for Systematic Reviews and Meta-Analyses (The PRISMA Statement) [29]. The review was prospectively registered on a Systematic Literature Review registration website (PROSPERO, Registration No. CRD42017077875). Research literature databases Ovid MEDLINE, Scopus, CINAHL, Embase and Web of Science were searched between 1 January 1982 and 31 December 2018. As the term “resistant starch” was not defined in the scientific literature until 1982 [30], databases were not searched prior to this year. Studies were included if published in English and involved human participants. All databases were searched using the terms (adult * OR patient * OR human) AND (‘resistant starch’ OR ‘retrograded starch’ OR ‘high amylose starch’ OR ‘RS2’ OR ‘high amylose adj2 starch’ OR ‘HAMSRS2’ OR ‘HAM-RS2’) AND (metabol * OR gluco * OR lipid OR ‘insulin resistan *’ OR appetite OR weight OR satiety OR ‘waist circumference’). The search strategy is presented in Appendix A. Studies with an RCT study design, recruited healthy human subjects or individuals with MetS or T2DM, provided 8 g or more of RS (type 2) per day to participants, and measured either fasting blood glucose, glycated hemoglobin (HbA1c), insulin resistance, appetite/satiety levels, lipid levels or body weight were eligible for inclusion. Excluded trials were not RCTs, supplemented less than 8 g of RS per day, provided RS1/RS3/RS4, involved participants with pre-existing gastrointestinal issues or terminal medical conditions, or measured non-metabolic outcomes. Reference lists of selected studies and reviews were manually searched to supplement the electronic search. 

#### 1.1.2. Screening and Eligibility

All resultant references were imported into a systematic review screening and data extraction software program (Covidence Systematic Review Software, Veritas Health Innovation, Melbourne, Australia), which was used to screen studies and identify those meeting the pre-specified inclusion criteria. The Covidence program automatically identified and eliminated duplicate articles. During the first pass, article titles and abstracts were screened by two of the listed authors (JJ, DM, SK, AL, RS) independently to determine their suitability for inclusion. Selected studies then underwent full-text screening, which was also conducted by two of the listed authors (JJ, DM, SK, AL, RS) independently. Conflicts were resolved by either a third author (NJK) or by discussion until consensus was reached. On completion of screening, the PRISMA Flowchart was automatically generated by the Covidence program.

#### 1.1.3. Risk of Bias Assessment

The risk of bias of eligible studies was independently assessed by two separate authors (NJK and JJ) using The Cochrane Risk of Bias tool for quality assessment of RCTs [31]. This tool identifies potential sources of bias within studies based on their use of the following bias minimization items: random sequence generation, allocation concealment, blinding of participants and personnel, blinding of outcome assessors, completeness of outcome data, non-selective outcome reporting, and other items which attempt to minimize bias. Authors assessing each criterion indicated whether the study satisfied each bias minimization item by recording “yes”, “no” or “unsure”. Studies that were assigned a “yes” for most of the items were considered to have a low risk of bias, whereas studies that were assigned a “no” or “unsure” for the majority of the items were considered to have either a moderate or high risk of bias. Inconsistencies between the authors risk of bias assessments at the study level were resolved through active discussion with the first assessor of the study until consensus was reached.

#### 1.1.4. Data Extraction and Statistical Methods

Upon completion of screening and risk of bias assessments, data were independently extracted from each article by all authors using a data collection table. Data collected included first author, year of publication, country in which the trial was conducted, mean age of participants (years), gender of participants (female or male), health status of participants (healthy, overweight or obese, diagnosed with MetS or T2DM), mean BMI of participants (kg/m^2^), dose of the RS2 (g/day), length of the intervention (weeks) and the effect of the RS2 intervention on metabolic outcomes. The following metabolic outcomes were extracted from each article: fasting plasma glucose (mmol/L), HbA1c (%), body weight (kg), insulin resistance (Homeostatic Model Assessment of Insulin Resistance (HOMA-IR)) or S (%), total cholesterol (mmol/L), LDL cholesterol (mmol/L), HDL cholesterol (mmol/L), triacylglycerol (TAG) (mmol/L) and subjective appetite/satiety ratings. These outcomes were subjected to random effects model meta-analyses using Review Manager (RevMan) (Computer program). Version 5.1. Copenhagen: The Nordic Cochrane Centre, The Cochrane Collaboration, 2014, Danmark. Using end-point data for control and intervention groups, the Mean differences (MDs) were determined for each outcome with 95% confidence intervals (CIs). Results were combined for each metabolic outcome and data were tested for interstudy heterogeneity using the Cochrane *Q* statistic and quantified by the *I*^2^ statistic with *P* < 0.10. An *I*^2^ ≥ 50% was considered substantial heterogeneity. Sensitivity analysis was conducted by omitting one study at a time to investigate the influence of a single study on each outcome estimate and heterogeneity. Where unexplained interstudy heterogeneity was identified, a random effects meta-regression analysis was undertaken in order to identify pre-defined covariates (participant age, BMI, health status, length of RS intervention or RS dose) which may contribute to the variation in effect sizes observed. Meta-regression analyses were undertaken using an appropriate software program (Comprehensive Meta-Analysis Version 3, Biostat, Englewood, NJ, USA, 2013). Publication bias was assessed by calculation of Egger’s regression asymmetry test and Begg’s test [32,33], with *P* < 0.05 considered evidence of small-study effects. For outcomes with ≥ten studies, funnel plots were constructed and visually assessed for funnel plot asymmetry. For the purpose of this meta-analysis, trial results were divided into different subgroups: healthy, overweight/obese, MetS (including prediabetes) and T2DM.

## 2. Results

### 2.1. Description of Selected Trials

Initial database searches yielded a total of 19,706 citations. Following the removal of duplicate articles and trials that did not meet the inclusion criteria, 22 RCTs [34,35,36,37,38,39,40,41,42,43,44,45,46,47,48,49,50,51,52,53,54,55] were available for qualitative analysis, and 20 RCTs [34,35,36,37,38,39,40,41,43,44,45,46,47,48,49,51,52,53,54,55] were available for quantitative analysis (Figure 1).

Characteristics of the included studies are detailed in Table 1. Articles were published from 1989 to 2018. A total of 670 participants, ranging from 23 to 70 years of age, were investigated in the 22 included trials. Subjects were either healthy, overweight/obese, or had been diagnosed with MetS, prediabetes or T2DM. Participant BMI ranged from 22 to 38 kg/m^2^. Trials randomized subjects to consume RS2 supplements or placebo for 1–12 weeks duration. Studies provided between 8 and 66 g of RS2 per day to participants. RS2 was primarily derived from high amylose maize starch, but one study [38] provided RS2 in the form of native banana starch and another study provided RS2 derived from potato starch [34]. From the 22 included studies, relevant outcomes reported included levels of fasting plasma glucose, HbA1c, HOMA-IR, total cholesterol, LDL cholesterol, HDL cholesterol, TAG, subjective appetite or satiety ratings and body weight. No studies reported any adverse participant side-effects following RS2 consumption.

### 2.2. Study Risk of Bias

The majority of studies included in the current review had an unclear risk of bias, with three studies demonstrating a high risk of bias and six studies demonstrating a low risk, as evaluated by the Cochrane Collaboration Risk of Bias tool (Table 2). Thirteen studies did not specify methods of allocation concealment or had not registered their primary outcome measures on a clinical trials database before trial commencement, and nine did not declare presence or absence of conflicts of interest. Sixteen studies did not blind outcome assessors. It should be noted that the CONSORT (Consolidated Standards of Reporting Trials) Statement, which outlines clear reporting guidelines, was first published in 1996 [56]. A number of studies included in this review were published prior to 1996, and the majority were published prior to 2010, before publication of the latest CONSORT Statement update [57]. This may account for the unclear risk of bias observed across the trials.

### 2.3. Fasting Plasma Glucose 

Fifteen studies [34,35,37,39,40,41,43,44,48,49,51,52,53,54,55] investigated the effect of RS2 on fasting plasma glucose in healthy subjects, subjects with MetS/prediabetes, and subjects with T2DM (Figure 2). The studies that reported on fasting plasma glucose showed no evidence of publication bias by Egger’s test (*P* = 0.620), Begg’s test (*P* = 0.268) or visual assessment of funnel plot (Appendix A). Sensitivity analysis indicated that none of the studies had a substantial influence on the combined results, with a range of −0.05 (95% CI −0.13, 0.03) to −0.02 mmol/L (95% CI −0.11, 0.07). No significant reductions in fasting plasma glucose were observed in any of the studies following daily consumption of RS2, when compared to placebo. A meta-analysis of five studies [34,35,37,44,54] conducted in 156 healthy individuals found no significant change in fasting plasma glucose (MD = 0.03 mmol/L; 95% CI −0.12, 0.18, *P* = 0.67), and an analysis of eight studies (*n* = 330) [39,41,44,49,51,52,53,55] summarizing the effect of RS2 consumption on individuals with MetS showed that the results were also not significantly different to placebo (MD = −0.03 mmol/L; 95% CI −0.16, 0.09, *P* = 0.60). Similarly, meta-analysis of three studies (*n* = 150) [40,43,48] undertaken in people with T2DM found no effect of RS2 supplementation on fasting plasma glucose levels in comparison to placebo (MD = −0.30 mmol/L; 95% CI −0.69, 0.10, *P* = 0.14). A meta-analysis of all fifteen studies (*n* = 636) indicated no statistically significant changes in fasting plasma glucose following RS2 supplementation in healthy subjects, subjects with MetS, and subjects with T2DM (MD = −0.03 mmol/L; 95% CI −0.11, 0.05, *P* = 0.40). Interstudy heterogeneity was minimal (*I²* = 0%, *P* = 0.47).

### 2.4. HbA1c 

Five studies [38,40,43,48,53] explored the effects of RS2 supplementation on HbA1c in subjects with prediabetes or T2DM (Figure 3). There was no indication of publication bias as assessed by Egger’s test (*P* = 0.338) and Begg’s test (*P* = 0.312). Sensitivity analysis did not indicate excessive contribution by any of the included studies to the combined results, with results ranging from −0.14 (95% CI −0.34, 0.06) to −0.37% (95% CI −0.78, 0.04). Despite significant reductions in HbA1c observed in two studies undertaken by the same research group [43,48], a meta-analysis of all five studies (*n* = 265) indicated no statistically significant effects of RS2 supplementation on HbA1c in trial participants with prediabetes or T2DM (MD = −0.27%; 95% CI −0.57, 0.03, *P* = 0.08). Interstudy heterogeneity was significant (*I*^2^ = 53%, *P* = 0.07). Systematic removal of individual trials indicated the study by Karimi [48] was a major source of heterogeneity (from *I*^2^ = 53%, *P* = 0.07 to *I*^2^ = 19%, *P* = 0.30).

### 2.5. Body Weight 

Six studies [37,38,39,40,54,55] assessed the effects of RS2 on body weight among healthy, overweight, MetS, and T2DM subjects (Figure 4). One study [38] with a high risk of bias conducted in 56 individuals with T2DM who were provided with native banana starch (8 g/day RS2) revealed a significant reduction in body weight after four weeks (MD = −1.30 kg; 95% CI −2.42, −0.18, *P* < 0.01). Sensitivity analysis showed that this study by Ble-Castillo and colleagues [38] contributed substantially to the combined results; when this study was removed from the analysis, the outcome MD was 0.02 kg (95% CI −3.79, 3.84). Conversely, when any of the other studies were removed from the analysis, the MD ranged from −1.20 (95% CI −2.29, −0.13) to −1.23 kg (95% CI −2.32, −0.14), indicating that none of the other studies had a large effect on the combined results. Furthermore, there was an indication of publication bias according to Egger’s test (*P* = 0.008), but not Begg’s test (*P* = 0.287). When, the study by Ble-Castillo and colleagues [38] was removed from the analysis, this publication bias no longer existed (Egger’s test: *P* = 0.270, Begg’s test: *P* = 0.164). No significant effects of RS2 on body weight were found in the other five trials, and a trial of longer duration (12 weeks) in people with T2DM which provided a greater quantity of RS2 (40 g/day) from high-amylose maize starch showed no change in participant’s body weight [40]. Due to the large effect size found in the trial by Ble-Castillo and colleagues [38], this study was heavily weighted (92.1%) which influenced the results of the meta-analysis. A meta-analysis including all six studies (*n* = 216) summarizing the effect of RS2 supplementation on body weight in healthy and overweight individuals and those with MetS or T2DM indicated a statistically significant reduction in body weight compared to placebo (MD = −1.19 kg; 95% CI −2.27, −0.12, *P* = 0.03). Statistical heterogeneity between studies was minimal (*I²* = 0%, *P* = 0.99). Subgroup analysis indicated that a significant reduction in body weight was only observed in individuals with T2DM (MD = −1.29 kg; 95% CI −2.40, −0.17, *P* = 0.02, *n* = 90), due to the effect size of the Ble-Castillo trial [38].

### 2.6. HOMA-IR

Twelve studies [34,38,39,40,41,44,47,48,49,52,54,55] examined the effect of RS2 supplementation on insulin resistance in healthy subjects and individuals with MetS or T2DM (Figure 5). There was no indication of publication bias by funnel plot (Appendix A), Egger’s regression asymmetry test (*P* = 0.542) or Begg’s test (*P* = 0.136). Sensitivity analysis indicated that no particular study had a significant contribution to the overall outcome estimate, with a range of −0.22 (95% CI −0.51, 0.07) to −0.11 (95% CI −0.39, 0.17). Two studies, both of eight weeks duration, reported significant reductions in insulin resistance following 25 g/day and 10 g/day RS2 supplementation in people with MetS (*n* = 47) [41] and T2DM (*n* = 56) [48], respectively, when compared to control groups. The remaining nine studies found no significant changes in insulin resistance following RS2 supplementation. Subgroup analyses indicated no changes in insulin resistance in healthy individuals (*n* = 62), those with MetS (*n* = 245) or in people with T2DM (*n* = 146) (Figure 5). A meta-analysis of all twelve studies (*n* = 453) found no significant effect of RS2 supplementation on insulin resistance when results from healthy subjects and those with MetS andT2DM were combined (MD = −0.17; 95% CI −0.46, 0.12, *P* = 0.25). Statistical heterogeneity between studies was significant (*I*^2^ = 43%, *P* = 0.05), but was reduced when the study by Karimi and colleagues [48] was excluded from analysis (*I*^2^ = 12%, *P* = 0.33). Meta-regression analysis was unable to identify any covariates (participant age, BMI, health status, length of RS intervention or RS dose) significantly associated with the variation in effect sizes observed (Appendix A). When a meta-analysis of the effects of RS2 on insulin sensitivity was performed using only studies that employed the use of the gold-standard euglycemic-hyperinsulinemic clamp to assess insulin sensitivity [40,47,54,55], the combined mean difference between intervention and placebo groups was also not statistically significant (data not shown).

### 2.7. Total Cholesterol 

Twelve studies [34,35,36,37,38,39,43,44,45,51,53,55] investigated the effect of RS2 supplementation on total cholesterol levels in healthy subjects, subjects with MetS, and subjects with T2DM (Figure 6). Sensitivity analysis ranged from −0.08 (95% CI −0.17, 0.01) to −0.02 mmol/L (95% CI −0.14, 0.10), indicating that no studies inordinately contributed to the results. Neither Egger’s test (*P* = 0.322), Begg’s test (*P* = 0.260) nor visual assessment of the funnel plot (Appendix A) provided any evidence for publication bias. Only one of these studies reported significant reductions in total cholesterol levels in people with T2DM (*n* = 60) [43] receiving dietary RS2 supplements in comparison to those receiving placebo. A meta-analysis of all twelve studies (*n* = 634) investigating the effect of RS2 supplementation on total cholesterol in healthy individuals and those with MetS or T2DM indicated no significant change in total cholesterol (MD = −0.06 mmol/L; 95% CI: −0.15, 0.03, *P* = 0.18). Statistical heterogeneity between studies was low (*I*^2^ = 0%, *P* = 0.45).

### 2.8. LDL Cholesterol 

Seven studies [34,37,40,43,45,51,53] evaluated the effect of RS2 supplementation on LDL cholesterol in healthy individuals (*n* = 207), subjects with MetS (*n* = 105), and those with T2DM (*n* = 94) (Figure 7). There was no indication of publication bias (Egger’s test: *P* = 0.357, Begg’s test: *P* = 0.326) and none of the studies had an excessive contribution to the combined results, with sensitivity analysis having a range from 0.01 (95% CI −0.14, 0.15) to 0.06 mmol/L (95% CI −0.08, 0.19). Subgroup meta-analyses indicated no significant change in LDL cholesterol levels in healthy subjects (MD = 0.03 mmol/L; 95% CI −0.15, 0.21, *P* = 0.74), subjects with MetS (MD = 0.08 mmol/L; 95% CI −0.20, 0.35, *P* = 0.59), and subjects with T2DM (MD = −0.12 mmol/L; 95% CI −0.65, 0.41, *P* = 0.65). A meta-analysis of all seven studies (*n* = 406) indicated no significant reduction in LDL cholesterol levels following RS2 supplementation (MD = 0.03 mmol/L; 95% CI −0.10, 0.16, *P* = 0.66). Interstudy heterogeneity was low (I^2^ = 0%, *P* = 0.64).

### 2.9. HDL Cholesterol 

Ten studies [34,35,37,38,40,43,44,45,51,53] measured the effect of RS2 supplementation on HDL cholesterol in healthy subjects, subjects with MetS, and subjects with T2DM (Figure 8). Nine of these studies reported no significant increase in HDL cholesterol following RS2 consumption. None of the studies made an excessive contribution to the combined results, with sensitivity analysis ranging from −0.04 (59% CI −0.08, 0.01) to 0.00 mmol/L (95% CI: −0.05, 0.05). There was no indication of publication bias according to Begg’s test (*P* = 0.349), Egger’s test (*P* = 0.435) or visual assessment of funnel plot (Appendix A). Gargari and colleagues [43] reported a significant increase in HDL cholesterol levels in people with T2DM following only 10 g/day RS2 supplementation for 8 weeks. However, a meta-analysis of the three studies investigating the effect of RS2 supplementation on HDL cholesterol levels in subjects with T2DM (*n* = 150) indicated no significant change in HDL cholesterol levels (MD = 0.06 mmol/L; 95% CI −0.11, 0.23, *P* = 0.50). A meta-analysis of all ten studies (*n* = 532) indicated no significant change in HDL cholesterol levels following RS2 supplementation in healthy people, those with MetS or individuals with T2DM (MD = −0.01 mmol/L; 95% CI −0.07, 0.04, *P* = 0.67). Heterogeneity between studies was moderate (*I*^2^ = 43%, *P* = 0.06). Systematic removal of individual trials found that the trial by Gargari et al. [43] was the source of all heterogeneity in the analysis (from *I*^2^ = 43%, *P* = 0.06 to *I*^2^ = 0%, *P* = 0.83).

### 2.10. Triacylglycerol (TAG)

Twelve studies [34,35,36,37,38,39,43,44,45,51,53,55] explored the effect of RS2 supplementation on serum TAG levels in healthy subjects and those with MetS or T2DM (Figure 9). Publication bias was excluded by visual assessment of the funnel plot (Appendix A), Egger’s test (*P* = 0.102) or Begg’s test (*P* = 0.260). Behall and colleagues [35] reported a significant reduction in TAG in 12 healthy subjects consuming 40 g/day RS2 for five weeks (MD = −0.20 mmol/L; 95% CI −0.34, −0.06, *P* = 0.013). A subgroup analysis of all six studies [34,35,36,37,44,45] investigating the effect of RS2 consumption on TAG in healthy subjects (*n* = 269) revealed a significant reduction in TAG levels (MD = −0.10 mmol/L; 95% CI −0.19, −0.01, *P* = 0.03). Heterogeneity within this subgroup was low (I² = 29%, *P* = 0.22). When the Behall study [35] was excluded from the analysis, the significant TAG reduction in the healthy subgroup disappeared (−0.07 mmol/L; 95% CI −0.16, 0.02, *P* = 0.14). Trials undertaken by Behall [36] and Gargari [43] in participants with MetS (*n* = 14) and T2DM (*n* = 60), respectively, also found significant reductions in TAG following RS2 consumption, but subgroup analyses conducted on all MetS trials and all T2DM trials were not significant and highly heterogeneous. A meta-analysis of all twelve studies (*n* = 634) indicated no significant change in TAG levels following RS2 supplementation (MD = −0.07 mmol/L; 95% CI −0.20, 0.05, *P* = 0.24). Sensitivity analysis showed that none of the studies excessively contributed to the meta-analysis results, with a range from −0.10 (95% CI: −0.22, 0.03) to −0.04 mmol/L (95% CI: −0.15, 0.08). Large interstudy heterogeneity was observed (*I*^2^ = 80%, *P* < 0.001), and when individual trials were systematically removed the overall heterogeneity remained substantial. Meta-regression analysis identified participant age, BMI and duration of RS intervention (time) as significant contributors to the variation in effect sizes observed. These three covariates combined, explained 57% of the variation in effect size (Appendix A).

### 2.11. Appetite

Three studies [42,46,50] investigated the effects of RS2 supplementation on subjective appetite and satiety ratings in either healthy [42,46] or overweight/obese [50] subjects, with participants reporting feelings of hunger and satiety on a visual analog scale (VAS). Jenkins and colleagues [46] reported a significant increase in satiety following 2-week consumption of the RS2 supplement compared to the low-fiber control, but there were no significant differences between satiety ratings reported after RS2 supplementation compared to a high-fiber (wheat bran) control (*n* = 24). DeRoos et al. [42] and Maziarz et al. [50] reported no significant changes in subjective appetite or satiety levels following RS2 supplementation. A meta-analysis was not performed due to insufficient data.

## 3. Discussion

Lifestyle changes, particularly dietary modification strategies, are recommended as the primary method of chronic disease prevention and management, given the convincing evidence supporting weight reduction and physical activity for the improvement of metabolic perturbations [58,59]. As the SCFA by-products of bacterial RS2 digestion have been proposed to confer beneficial effects on host glucose, insulin and lipid regulation, dietary RS2 supplementation may be one such dietary modification that could potentially improve cardiometabolic health [60]. This systematic review aimed to summarize the scientific literature regarding the effects of RS2 supplementation on body weight, appetite, glucose and lipid homeostasis in healthy adult individuals or those with MetS or T2DM. Evaluation of 22 RCTs involving 670 adults found that 1−12 weeks RS2 supplementation resulted in a significant reduction in TAG concentrations in healthy individuals, and a reduction in body weight among people with T2DM, but no significant changes in any other cardiometabolic variables were identified following meta-analyses. It should also be noted that the significant outcomes for TAG and body weight were strongly influenced by the results of a small number of studies, whose findings contradicted those of the majority of other trials.

Resistant starches are long chain, soluble, non-viscous fibers, so they are unable to form gels in the gastrointestinal tract or slow gastric emptying. Rather, their mechanism of action is related to their bacterial fermentation in the large intestine. The role of RS as a potential prebiotic may underlie its suggested physiological effects [61]. Prebiotics are defined as substrates that are selectively fermented by specific gut microbiota, providing a health benefit to the host [62]. This process of fermentation encourages the proliferation of SCFA-producing colonic bacteria [63]. Increased SCFA concentrations in the large intestine lowers the pH of the colonic environment, thereby preventing the growth of pathogenic bacterial species [64]. SCFAs have also been found to upregulate the expression of gut peptides such as glucagon-like peptide 1 (GLP-1) and peptide YY (PYY), compounds implicated in the regulation of appetite, glucose and insulin homeostasis [65]. GLP-1 increases satiety by inhibiting glucagon secretion, delaying gastric emptying and acting centrally by interacting with appetite regulating centers within the hypothalamus and the brainstem [66,67,68]. Circulating PYY also acts on appetite regulating centers such as the arcuate nucleus within the hypothalamus to increase anorexigenic neuronal activity and inhibit orexigenic neuronal activity, thereby increasing satiety [69].

Studies in rats and mice consuming RS show consistent increases in the expression of PYY, GLP-1 and adiponectin along with enhanced fatty acid oxidation in the liver [70,71]. Rodents consuming RS demonstrate reductions in abdominal fat [72], increased fat oxidation [73] and improvements in insulin sensitivity [74]. Despite this, the results of human intervention trials have not replicated the convincing health outcomes seen in rodent RS studies, likely due to the lower quantities of RS tolerated by humans, genetic heterogeneity, the inter-individual diversity of the existing gut microbiota and the influence of diet, exercise, cigarette smoking, alcohol consumption, medical conditions, sleep patterns and medication use on human metabolism [75]. A number of rodent and human studies have also failed to match the metabolizable energy content of RS (8−10 kJ/g) with the rapidly digestible starch control (17 kJ/g) [17], resulting in the intervention group receiving a treatment with lower energy density and available carbohydrate content, potentially influencing body weight and metabolic outcomes.

Most trials investigating the effect of RS2 supplementation on body weight in humans have found the fiber to be weight neutral. This review found insufficient evidence to support the consumption of RS2 for the reduction of body weight or appetite in healthy and overweight individuals, or those with the metabolic syndrome. Ble-Castillo [38] reported a significant weight reduction in adults with T2DM following the consumption of 8 g/day of RS2 from native banana starch for four weeks, but this study was determined to have a high risk of bias. In contrast, Bodinham and colleagues [40] found no change in the mean body weight of people with T2DM who consumed 40 g/day RS2 from high amylose maize starch for 12 weeks. The risk of bias in this study was unclear. While it would be expected that RS2 derived from banana starch and RS2 derived from high amylose maize starch are structurally identical, we cannot rule out the possibility that the inconsistencies in weight loss between the two trials were due to unique physiological effects elicited by the banana starch or the presence of other components within the starch matrix. Dodevska and colleagues [20] reported small but significant body weight reductions in overweight and obese individuals with prediabetes following RS2 supplementation, but this intervention was provided in conjunction with a healthy lifestyle program so did not meet the criteria for inclusion in this review. Nichenametla et al. [76] found a significant reduction in waist circumference, but not body weight, in healthy individuals consuming RS4, but the properties and health effects of RS4 cannot be generalised to RS2 [77]. Based on current evidence, RS2 consumption from high amylose maize starch is unlikely to induce reductions in appetite and body weight of a magnitude considered clinically significant in human populations.

A healthy gastrointestinal epithelium is important in preventing the translocation of bacterial lipopolysaccharide (LPS) from the gut into the bloodstream, which is hypothesized to increase immune-stimulated oxidative stress and induce insulin resistance [78,79]. Butyrate, a SCFA produced by RS2-fermenting gut bacteria, plays a significant role in maintaining the health and integrity of the gastrointestinal barrier [80]. Butyrate is an essential fuel source for healthy colonocytes, which up-regulates GLP-2 expression, a hormone which promotes the proliferation of new gastrointestinal cells and increases the production of tight-junction proteins which modulate gut permeability [81]. Butyrate also enhances the expression of nuclear receptor PPAR-γ, which plays a role in attenuating gastrointestinal inflammation, and upregulates the mucin-associated genes responsible for generating the production of the thick mucin layer that plays an important role in maintaining the integrity of the intestinal mucosal barrier [82]. RS2 feeding in rodent studies improves glucose tolerance and decreases insulin resistance [25], but it is too early to apply these findings to humans. Some evidence indicates that RS2 supplements may need to be taken as part of a low-fat diet, to ensure sufficient quantities of specific RS-degrading bacteria are present in the large intestine [83]. Overall, this review found no significant impact of RS2 consumption on fasting glucose, glycated hemoglobin or insulin resistance in healthy subjects or people with MetS or T2DM.

The SCFA propionate is hypothesized to play a significant role in the modification of hepatic lipid metabolism. In the liver, propionate may contribute to the inhibition of cholesterol synthesis by reducing the activity of HMG-CoA reductase [84]. One study investigating the effect of adzuki bean RS on rats reported significantly decreased hepatic HMG-CoA expression following RS supplementation [85]. In addition, dietary RS supplementation might attenuate cholesterol and TAG production by stimulating the synthesis of cis−9, trans−11-conjugated linoleic acid (CLA) from polyunsaturated fatty acids by beneficial bacterial species such as *Bifidobacterium breve*. This CLA isoform has been shown to reduce cholesterol and triglyceride concentrations in animal studies [86], but the results of human trials are less conclusive. Gut microbes are also involved in the transformation of primary bile acids into secondary bile acids in the colon. These bile acids are deconjugated and are therefore unavailable for enterohepatic recirculation. As a result, the liver is forced to produce additional bile acids from circulating cholesterol [87]. However, primary bile acids are only available for reabsorption from the terminal ileum; so once they have entered the colon, they are unlikely to significantly influence hepatic bile acid production.

Despite positive results in animal studies where relatively high doses of RS were administered, most human RS2 supplementation trials included in this review failed to clearly demonstrate significant lipid-lowering effects compared to placebo. A meta-analysis of six studies [34,35,36,37,44,45] investigating the effect of RS2 consumption on TAG in healthy subjects (*n* = 269) revealed a significant reduction in TAG levels. However, it should be noted that many RS2 intervention trials were of limited duration and are not sufficient to determine the long-term impact of RS on circulating lipids. In summary, RS2 consumption does not appear to effect lipid concentrations in humans when consumed in realistic quantities.

There are some limitations of this review. Most of the studies included in this review were determined to have an uncertain risk of bias (Table 2). For studies published prior to 2010, high or uncertain risk of bias may be more related to the lack of reporting according to the requirements of the CONSORT Statement rather than bias inherent in the study design. Additionally, the trials by Gargari [43] and Karimi [48] appear to have used the same study participants. The same authors contributed to the two papers, despite describing two unique studies. Further, the large effect sizes reported in these trials have not been replicated by other research groups in people with T2DM. Out of the 22 different studies in this review, only six had a low risk of bias, making it difficult to interpret results with confidence. Many of the appraised studies had small sample sizes, which may not be representative of the true population and their characteristics. One meta-analysis (Figure 9) displayed high levels of statistical heterogeneity between studies, making it more difficult to make comparisons and draw conclusions. Brief treatment durations (1−4 weeks) using small doses of RS2 (<20 g/day) may have been insufficient to generate significant outcomes. Further research is required to determine the optimal daily dose and duration of treatment required in order to maximize health benefits. RS may even confer metabolic benefits independent of the gut microbiota, with one research group demonstrating improvements in insulin sensitivity in both germ-free and conventionalized mice fed RS. The authors hypothesized that RS may modulate the cecal bile acid pool, reducing macrophage migration in adipose tissue [88]. 

While large quantities of RS have successfully reduced cardiometabolic risk factors and improved glycemic control in animal studies, human trials have been unable to clearly demonstrate beneficial effects of dietary RS2 consumption on appetite, body weight, fasting blood glucose, insulin resistance or cholesterol concentrations when consumed in realistic doses. Bacterial production of SCFAs from RS is a dose-dependent phenomenon, and RS often contributes 30−55% to the diet of laboratory animals [23], which may explain the discrepancies in outcomes between human and animal studies. In order to replicate the beneficial metabolic outcomes observed in animal studies in human subjects, it may be necessary to administer SCFAs directly into the diet, gut or circulation.

This systematic literature review found limited evidence to support RS2 supplementation for the improvement of cardiometabolic outcomes in healthy individuals or those with MetS or T2DM. Even when considering the small number of individual trials reporting significant results, the effect sizes observed were of minimal clinical significance. Larger studies with lower risk of bias and longer duration should be undertaken to confirm these findings. 

## Figures and Tables

**Figure 1 nutrients-11-01833-f001:**
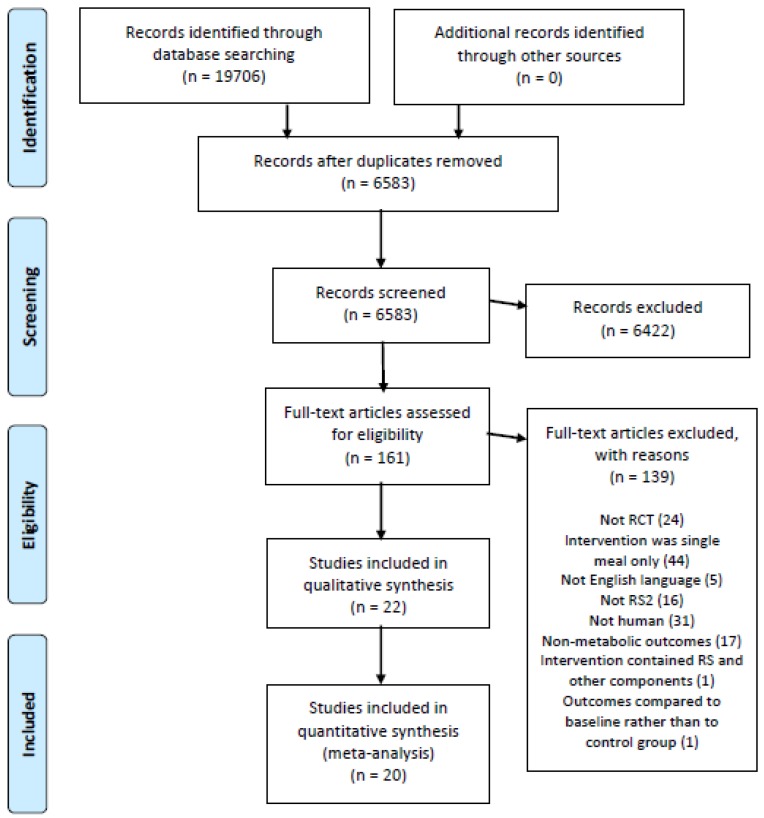
Literature search and review flowchart for selection of studies.

**Figure 2 nutrients-11-01833-f002:**
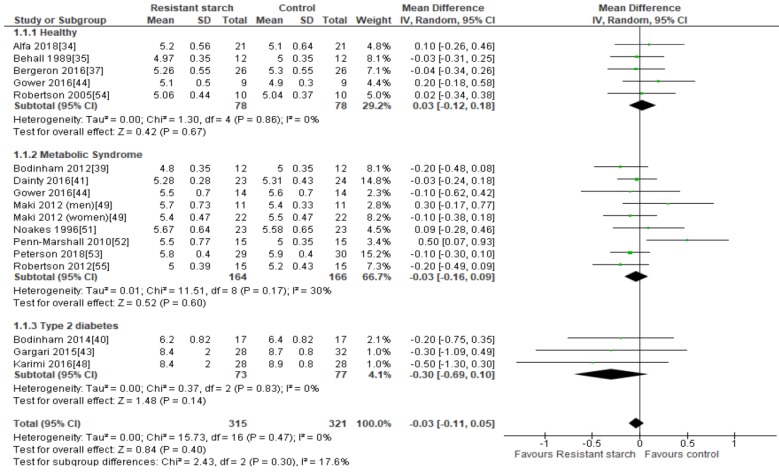
Effect of RS2 supplementation on fasting plasma glucose (mmol/L) in healthy subjects, subjects with MetS and subjects with T2DM. Mean Difference (95% CI) shown for individual and pooled trials.

**Figure 3 nutrients-11-01833-f003:**
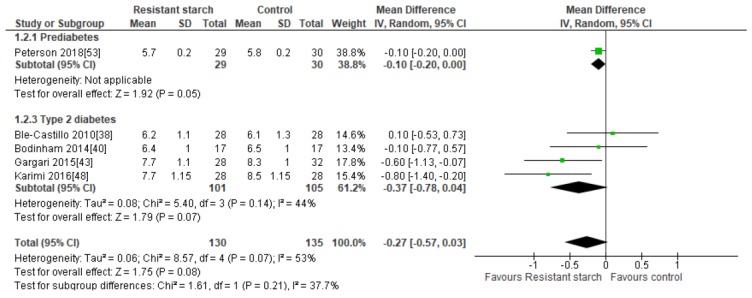
Effect of RS2 supplementation on glycated hemoglobin (HbA1c %) in subjects with T2DM. Mean Difference (95% CI) shown for individual and pooled trials.

**Figure 4 nutrients-11-01833-f004:**
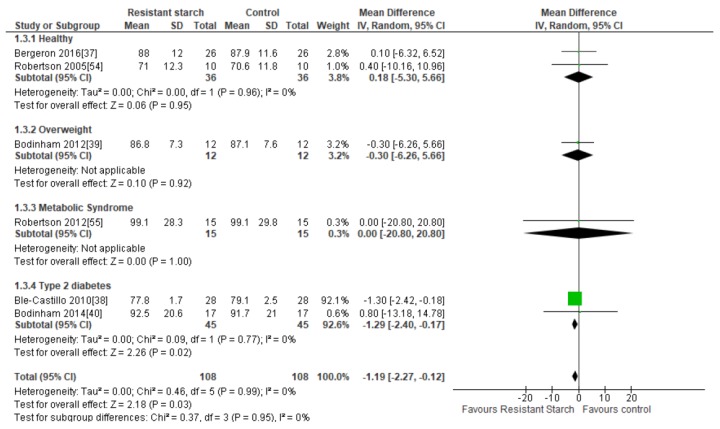
Effect of RS2 supplementation on body weight (kg) in healthy subjects, overweight subjects, subjects with MetS, and subjects with T2DM. Mean Difference (95% CI) shown for individual and pooled trials.

**Figure 5 nutrients-11-01833-f005:**
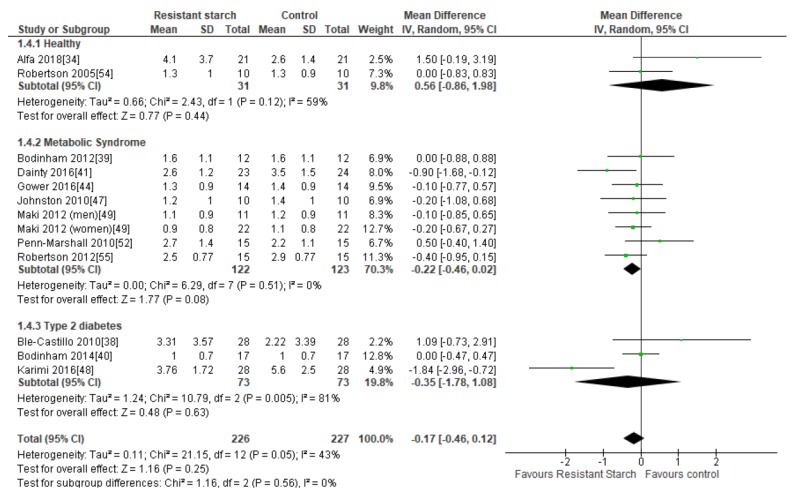
Effect of RS2 supplementation on HOMA-IR in healthy subjects, subjects with MetS and subjects with T2DM. Mean Difference (95% CI) shown for individual and pooled trials.

**Figure 6 nutrients-11-01833-f006:**
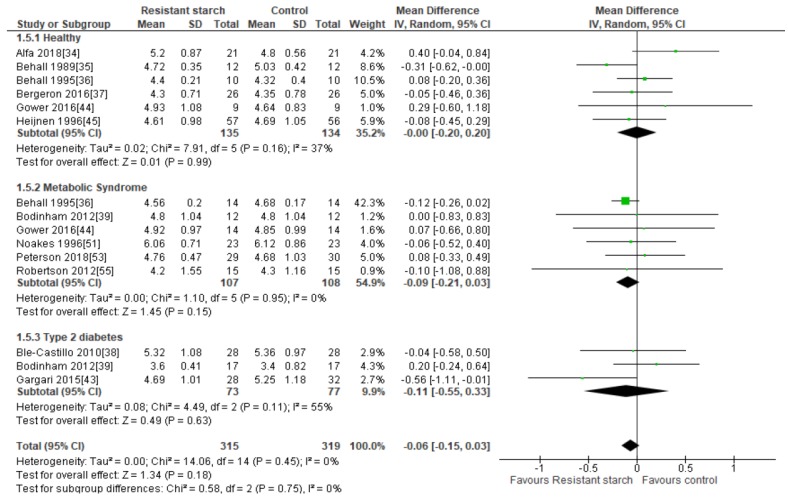
Effect of RS2 supplementation on total cholesterol (mmol/L) in healthy subjects, subjects with MetS and subjects with T2DM. Mean Difference (95% CI) shown for individual and pooled trials.

**Figure 7 nutrients-11-01833-f007:**
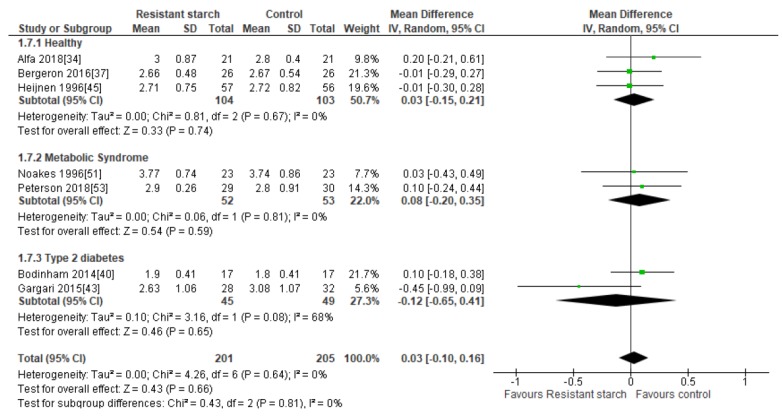
Effect of RS2 supplementation on LDL cholesterol (mmol/L) in healthy subjects, subjects with MetS and subjects with T2DM. Mean Difference (95% CI) shown for individual and pooled trials.

**Figure 8 nutrients-11-01833-f008:**
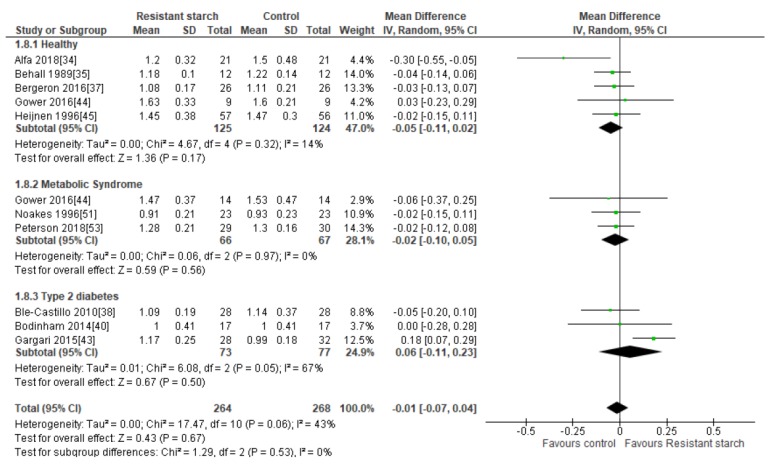
Effect of RS2 supplementation on HDL cholesterol (mmol/L) in healthy subjects, subjects with MetS and subjects with T2DM. Mean Difference (95% CI) shown for individual and pooled trials.

**Figure 9 nutrients-11-01833-f009:**
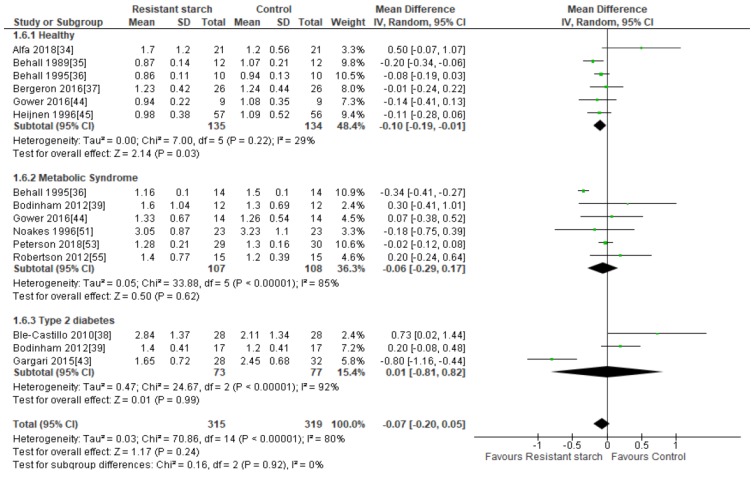
Effect of RS2 supplementation on TAG (mmol/L) in healthy subjects, subjects with MetS and subjects with T2DM. Mean Difference (95% CI) shown for individual and pooled trials.

**Table 1 nutrients-11-01833-t001:** Characteristics of included studies.

Study Author/Year	Participants (Country, Age, No. of Participants, Gender, Health Status, BMI)	Study Design	Resistant Starch Intervention (Dose)	Type of RS	Length of Intervention	Effect of RS Intervention Compared to Placebo on Metabolic Outcomes
Alfa et al. (2018) [34]	Canada, *n* = 42 healthy adults (24 females, 18 males), age range 32–50 years; median body weight 78.4 kg	Parallel RCT, double blinded	Random assignment to 2 groups (control: fully digestible corn starch, intervention: 70% resistant potato starch) RS = 30 g/day supplement containing 21 g RS2	RS2 (*Solanum tuberosum* extract)	12 weeks	↔ fasting glucose ↔ total cholesterol ↔ HDL ↔ LDL ↔ TAG ↔ HOMA-IR
Behall et al. (1989) [35]	USA, *n* = 12 healthy men (mean age 34 years, mean body weight 77.3 kg)	Crossover RCT, blinding not specified	Random assignment to 2 groups (control: 70% amylopectin corn starch, intervention: 70% amylose corn-starch), no washout period RS = approx. 40 g/day	RS2 (HAM-RS2)	5 weeks	↔ fasting glucose ↓ total cholesterol ↔ HDL ↓ TAG
Behall et al. (1995) [36]	USA, *n* = 24 healthy or hyperinsulinemic subjects (mean age control 37.2 years, intervention 41.2 years; mean BMI control 24.2 kg/m^2^, intervention 27.1 kg/m^2^)	Crossover RCT, blinding not specified	Random assignment to 2 groups (control: 70% amylopectin + 30% amylose starch, intervention: 30% amylopectin + 70% amylose resistant dextrin), 4-week washout RS ≥ 55% of total CHO intake/day	RS2 (HAM-RS2)	10 weeks	↔ total cholesterol (healthy) ↔ total cholesterol (MetS) ↔ TAG (healthy) ↓ TAG (MetS)
Bergeron et al. (2016) [37]	USA, *n* = 52 healthy men and women (mean age 44 years, mean BMI 31 kg/m^2^)	Crossover RCT, double blinded	Random assignment to 4 groups (high CHO/high RS, high CHO/low RS, low CHO/high RS, low CHO/low RS), 2-week washout RS = 48–66 g/day	RS2 (HAM-RS2)	2 weeks	↔ fasting glucose ↔ total cholesterol ↔ HDL ↔ LDL ↔ TAG ↔ body weight
Ble-Castillo et al. (2010) [38]	Mexico, *n* = 30 adults with T2D (mean age 51.7 years; mean BMI 34.89 kg/m^2^)	Crossover RCT, single blinded	Random assignment to 2 groups (control: soy drink, intervention: native banana starch NBS drink), no washout period RS = 24 g/day NBS contained 8 g RS2	RS2 (Native Banana Starch)	4 weeks	↔ fasting glucose ↔ insulin resistance ↔ HBA1c ↔ total cholesterol ↔ LDL ↔ HDL ↔ TAG ↓ body weight
Bodinham et al. (2012) [39]	UK, *n* = 12, overweight participants (mean age 37 years; mean BMI control 28.4 kg/m^2^, intervention 28.4 kg/m^2^)	Crossover RCT, single blinded	Random assignment to 2 groups (control: 27 g rapidly digestible starch or intervention: 67 g Hi-maize 260), 4-week washout RS = 40 g/day	RS2 (HAM-RS2)	4 weeks	↓ fasting glucose ↔ Insulin sensitivity ↑ fasting insulin ↑ first-phase insulin secretion ↔ total cholesterol ↔ HDL ↔ LDL ↔ TAG ↔ body weight ↔ waist circumference
Bodinham et al. (2014) [40]	UK, *n* = 17 adults with well-controlled T2DM (mean age 55 years; mean BMI 30.6 kg/m^2^)	Crossover RCT, single blinded	Random assignment to 2 groups (control: 27 g rapidly digestible starch or intervention: 67 g Hi-maize 260), 12-week washout RS = 40 g/day	RS2 (HAM-RS2)	12 weeks	↔ fasting glucose ↔ HbA1c ↔ insulin resistance (clamp) ↔ total cholesterol ↑ TAG ↔ weight ↔ waist circumference
Dainty et al. (2016) [41]	Canada, *n* = 24 adults with MetS (mean age 55.3, mean BMI 30.2 kg/m^2^)	Crossover RCT, double blinded	Random assignment to 2 groups (control: wheat bagel, intervention: high amylose maize RS bagel, 4-week washout RS = 25 g/day	RS2 (HAM-RS2)	8 weeks	↔ fasting glucose ↓ HOMA-IR ↔ body weight ↔ BMI
deRoos et al. (1995) [42]	The Netherlands, *n* = 24 healthy males (mean age 23; mean BMI 22.7 kg/m^2^)	Crossover RCT, single blinded	Random assignment to 3 groups (control: glucose, intervention 1: RS2, intervention 2: Retrograded RS3) RS = 30 g/day	RS2 and RS3 (only results for RS2 reported)	1 week	↔ satiety
Gargari et al. (2015) [43]	Iran, *n* = 60 women with T2DM (mean age control 49.6 years, intervention 49.5 years; mean BMI control 30.8 kg/m^2^, intervention 31.5 kg/m^2^)	Parallel RCT, triple blinded	Random assignment to 2 groups (control: maltodextrin, intervention: Hi Maize 260 RS2) RS = 10 g/day	RS2	8 weeks	↔ fasting glucose ↓ HbA1c ↔ total cholesterol ↔ LDL ↑ HDL ↓ TAG
Gower et al. (2016) [44]	USA, *n* = 40 healthy or insulin resistant women (mean age 48.3 years; mean BMI 29.8 kg/m^2^)	Crossover RCT, double blinded	Random assignment to 3 groups (control: waxy corn starch, intervention 1: low RS, intervention 2: high RS), 4-week washout Low RS = 15 g/day High RS = 30 g/day	RS2 (HAM-RS2) (only results for High RS reported)	4 weeks	↔ fasting glucose (healthy) ↔ fasting glucose (MetS) ↔ insulin sensitivity (healthy) ↑ insulin sensitivity (MetS) ↔ total cholesterol (healthy) ↔ total cholesterol (MetS) ↔ HDL (healthy) ↔ HDL (MetS) ↔ TAG (healthy) ↔ TAG (MetS)
Heijnen et al. (1996) [45]	The Netherlands, *n* = 60 healthy males and females (mean age 24.0 years; mean BMI 22.3 kg/m^2^)	Crossover RCT, single blinded	Random assignment to 6 groups. Each group consumed the following 3 supplements in one of 6 different sequences, with no washout period in between (control: glucose, intervention 1: RS2—High amylose resistant corn starch, intervention 2: RS3—Retrograded high amylose resistant corn starch) RS3 interventio*n* = 30 g/day RS2 interventio*n* = 30 g/day	RS2 and RS3 (only results for RS2 reported)	3 weeks	↔ total cholesterol ↔ HDL ↔ LDL ↔ TAG ↔ body weight
Jenkins et al. (1998) [46]	Canada, *n* = 24 healthy adults (mean age 33 years, mean BMI 23.7 kg/m²)	Crossover RCT, blinding not specified	Random assignment to 4 groups (control group 1: low fiber, control group 2: 30 g wheat bran, intervention 1: RS2—High amylose resistant starch, intervention 2: RS3—Retrograded high amylose resistant cornstarch), 2-week washout RS = 21.5 g/day	RS2 and RS3 (only results for RS2 reported)	2 weeks	↔ total cholesterol ↔ HDL ↔ LDL ↔ TAG ↑ satiety
Johnston et al. (2010) [47]	UK, *n* = 20 insulin resistant adults (8 females, 12 males; mean age control 50.1 years, mean age intervention 45.2 years; mean BMI control 30.4 kg/m^2^, mean BMI intervention 31.3 kg/m^2^)	Parallel RCT, single blinded	Random assignment to 2 groups (control: rapidly digestible starch, intervention: Hi-Maize 260 RS) RS = 40 g/day	RS2 (HAM-RS2)	12 weeks	↑ insulin sensitivity (clamp) ↔ body weight
Karimi et al. (2016) [48]	Iran, *n* = 56 women with T2DM (mean age control 48.6 years, mean age intervention 49.5 years; mean BMI control 31.0 kg/m^2^, mean BMI intervention 31.5 kg/m^2^)	Parallel RCT, triple blinded	Random assignment to 2 groups (control: maltodextrin, intervention: Hi-Maize 260 RS) RS = 10 g/day	RS2 (HAM-RS2)	8 weeks	↔ fasting glucose ↓ HbA1c ↓ HOMA-IR
Maki et al. (2012) [49]	USA, *n* = 33 overweight or obese adults (mean age 49.5 years; mean BMI 30.6 kg/m^2^)	Crossover RCT, double blinded	Random assignment to 3 groups (control: rapidly digestible starch, intervention 1: low HAM-RS2, intervention 2: high HAM-RS2), 3-week washout Low RS = 15 g/day High RS = 30 g/day	RS2 (HAM-RS2) (only results for High RS reported)	4 weeks	↔ fasting glucose ↔ insulin sensitivity (women) ↑ insulin sensitivity (men)
Maziarz et al. (2017) [50]	USA, *n* = 18 overweight adults (mean age control 31.2 years, intervention 31.0 years; mean BMI control 30.6 kg/m^2^, intervention 34.8 kg/m^2^)	Parallel RCT, double blinded	Random assignment to 2 groups (control: placebo muffin, intervention: muffin containing HAM-RS2) RS = 30 g/day	RS2 (HAM-RS2)	6 weeks	↔ satiety
Noakes et al. (1996) [51]	USA, *n* = 23 overweight adults with hypertriglyceridemia (mean age women 51 years, men 51 years; mean BMI women 29 kg/m^2^, men 29 kg/m^2^)	Crossover RCT, single blinded	Random assignment to 3 groups (control: low amylose, intervention 1: high oat bran, intervention 2: high amylose cornstarch (17 g RS for women and 25 g RS for men), no washout RS = 17 g RS for women, 25 g RS for men	RS2 (HAM-RS2)	4 weeks	↔ fasting glucose ↔ total cholesterol ↔ HDL ↔ LDL ↔ TAG
Penn-Marshall et al. (2010) [52]	USA, *n* = 15 adults with MetS (mean age 36.6 years; mean BMI 37.7 kg/m²)	Crossover RCT, double blinded	Random assignment to 2 groups (control: regular bread, intervention: bread made with added RS), 2-week washout RS = 12 g/day	RS2 (HAM-RS2)	6 weeks	↔ fasting glucose ↔ fasting insulin ↔ HOMA-IR ↔ fructosamine ↔ body weight ↔ BMI ↔ waist circumference
Peterson et al. (2018) [53]	USA, *n* = 59 overweight or obese adults with diagnosed prediabetes (20 males, 39 females); mean age 55 years; mean BMI 35.6 kg/m²	Parallel RCT, double blinded	Random assignment to 2 groups (control: rapidly digestible cornstarch amylopectin, intervention: Hi-Maize 260 RS) RS = 45 g/day	RS2 (HAM-RS2)	12 weeks	↓ HbA1c ↔ fasting glucose ↔ fasting insulin ↔ HOMA-IR ↔ total cholesterol ↔ HDL ↔ LDL ↔ TAG
Robertson et al. (2005) [54]	UK, *n* = 10 healthy adults (mean age 48.5 years; mean BMI 23.4 kg/m^2^)	Crossover RCT, single blinded	Random assignment to 2 groups (control: rapidly digestible starch, intervention: Hi-Maize 260), 4-week washout RS = 30 g/day	RS2 (HAM-RS2)	4 weeks	↔ fasting glucose ↔ HOMA-IR ↑ insulin sensitivity (clamp) ↔ body weight ↔ BMI
Robertson et al. (2012) [55]	UK, *n* = 15 adults with MetS (Age range 25–70 years; mean BMI 33.8 kg/m^2^)	Crossover RCT, single blinded	Random assignment to 2 groups (control: rapidly digestible starch, intervention—HAM-RS2), 8-week washout RS = 40 g/day	RS2 (HAM-RS2)	8 weeks	↔ fasting glucose ↔ insulin resistance (clamp) ↔ total cholesterol ↔ TAG ↔ body weight

BMI: Body Mass Index (kg/m^2^); T2DM: Type 2 diabetes mellitus; CHO: Carbohydrate; HAM-RS2: High Amylose Maize-Resistant Starch 2; HbA1c: Hemoglobin A1c (glycated hemoglobin); HDL: High-Density Lipoprotein; HOMA-IR: Homeostatic Model Assessment of Insulin Resistance; LDL: Low- Density Lipoprotein; MetS: Metabolic Syndrome; RCT: Randomized Controlled Trial; RS: Resistant Starch; TAG: Triacylglycerol; T2D: type 2 diabetes; ↓: significantly lower than that in the comparison control group after intervention; ↑: significantly higher than that in the comparison control group after intervention; ↔: no significant difference between the RS-supplemented and control groups after intervention.

**Table 2 nutrients-11-01833-t002:** Risk of bias summary for included studies.

Study Author/Year	Risk of Bias ^a^	Bias Minimization Items ^b^						
		1	2	3	4	5	6	Other
Alfa et al. (2018) [34]	Low	+	+	?	+	+	+	Unclear whether funding and sponsorship free from bias
Behall et al. (1989) [35]	Unclear	?	?	?	?	+	?	Unclear whether funding and sponsorship free from bias
Behall et al. (1995) [36]	High	?	?	?	?	+	-	Unclear whether funding and sponsorship free from bias
Bergeron et al. (2016) [37]	Low	+	+	+	-	?	+	Funding and sponsorship free from bias
Ble-Castillo et al. (2010) [38]	High	-	?	?	-	+	+	Funding and sponsorship free from bias
Bodinham et al. (2012) [39]	High	-	?	+	-	?	+	Funding and sponsorship free from bias
Bodinham et al. (2014) [40]	Unclear	+	?	+	-	?	?	Funding and sponsorship free from bias
Dainty et al. (2016) [41]	Unclear	+	+	?	?	+	?	Unclear whether funding and sponsorship free from bias
De Roos et al. (1995) [42]	Unclear	?	?	+	?	+	?	Unclear whether funding and sponsorship free from bias
Gargari et al. (2015) [43]	Unclear	+	+	+	+	+	?	Funding and sponsorship free from bias. Unspecified overlap between subjects participating in this study and another [48] undertaken by the same research group. Significant outcomes not replicated by other studies performed in similar patient groups.
Gower et al. (2016) [44]	Low	?	+	+	?	+	+	Unclear whether funding and sponsorship free from bias
Heijnen et al. (1996) [45]	Unclear	?	?	?	+	+	?	Funding and sponsorship free from bias
Jenkins et al. (1998) [46]	Unclear	?	?	?	?	+	?	Unclear whether funding and sponsorship free from bias
Johnston et al. (2010) [47]	Unclear	?	?	+	?	+	?	Funding and sponsorship free from bias
Karimi et al. (2016) [48]	Unclear	+	+	+	+	+	?	Funding and sponsorship free from bias. Unspecified overlap between subjects participating in this study and another [43] undertaken by the same research group. Significant outcomes not replicated by other studies performed in similar patient groups.
Maki et al. (2012) [49]	Unclear	-	?	+	+	?	+	Unclear whether funding and sponsorship free from bias
Maziarz et al. (2017) [50]	Low	+	+	+	+	+	+	Funding and sponsorship free from bias
Noakes et al. (1996) [51]	Unclear	?	?	+	?	?	?	Funding and sponsorship free from bias
Penn-Marshall et al. (2010) [52]	Unclear	?	?	+	?	+	+	Funding and sponsorship free from bias
Peterson et al. (2018) [53]	Low	+	+	+	+	+	+	Unclear whether funding and sponsorship free from bias
Robertson et al. (2005) [54]	Unclear	?	?	?	?	?	?	Funding and sponsorship free from bias
Robertson et al. (2012) [55]	Low	-	+	+	-	+	+	Unclear whether funding and sponsorship free from bias

“+” = response of “yes” to use of the bias minimization item; “-” = response of “no” to use of the bias minimization item; “?” = response of “uncertain” to the use of the bias minimization item; ^a^ Assessed using the Cochrane Collaboration tool for assessing risk of bias in RCTs [31]; ^b^ Bias minimization items: 1. Random sequence generation (selection bias); 2. Allocation concealment (selection bias); 3. Blinding of participants and personnel (performance bias); 4. Blinding of outcome assessment (detection bias); 5. Complete outcome data (attrition bias); 6. Non-selective reporting (reporting bias). Trials receiving a + response for most items are likely to have a low risk of bias.

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
