# Peer review of "Metabolic Effects of Resistant Starch Type 2: A Systematic Literature Review and Meta-Analysis of Randomized Controlled Trials"

_nutrients, 2019, doi:10.3390/nu11081833_

Round 1

Reviewer 1 Report

The manuscript of Snelson and colleagues based on meta-analysis of published randomised controlled trials on the effect of resistant starch type 2 supplementation on various metabolic factors in healthy, overweight, and people with metabolic syndrome including T2DM.

Although I declare no previous experience in the design of meta-analysis studies, this is a well conducted study in my opinion. However there are few major factors that are basically show stoppers.

Data do not support conclusion. In other words, the data in the abstract about serum lipid do not support the conclusion.  “This systematic literature review found some evidence to support a statistically significant reduction in circulating TAG concentrations following RS2 supplementation in healthy individuals” While only one study showed significant improving in in TG lipid profile in the healthy group  it would be wrong to give this general statement in the abstract and enforce it in the conclusion.

Have the authors tried to understand why data from this group are different?  Behall et al. (1985) recruited only men who are relatively young (34 years). The number of subjects is small (n=12).  Have the authors looked at the effect of those factors (gender, age) when interpreted and discussed the results?

The authors used the term short duration studies in the manuscript and they stated in the abstract: “Most studies had an unclear risk of bias, were of short  duration (1-12 weeks)”

Again, this is not very careful statement because all the studies that included in Table 1 were of 1-12 weeks (I was not able to identify studies of longer than 12 weeks). In addition, any long duration effect should be expected to be >12 months, and all studies that were included fall shorter than this classification.

Although the manuscript is well written, presentation of the data require improvement to help the reader understand the main concept. For examples:

Under Methods: Information is very dense making it hard for the reader to follow. The authors used nicely figure 1 to explain the process, it would be clearer to borrow the same heading from figure 1 when describing the methods on pages 3 and 4.

The title of Figure 1 is not clear: it would be better to be changed to something like: Flow diagram shows the selection process of the randomized trials.

The legend to this figure is not appropriate as well, the authors described methodological data that better inserted under methods. The legend should describe the data in the flow diagram only (i.e. number of participants at each stage and reason for exclusions).

Table 1: this contain very useful data, but too large to be included in the main body of the manuscript. I suggest this is lifted to the supplementary.

I am confused whether all figures from 2-9 are required? Is this the best way to present the data? Is this technically a graph or a table?

Minor issues

All the units should be removed from the headings under result section.

Examples

Line 198. Fasting Plasma Glucose: delete (mmol/L) from the heading

Line 234: Body Weight:  delete (Kg) from the heading

Line 123: article titles and abstracts were screened by two authors (JJ, DM, SK, AL, RS): not clear?

Line 125: which was also conducted by two authors (JJ, DM, SK, AL, RS): not clear?

Reviewer 2 Report

In this review the authors aim to summarize the effects of dietary RS2 supplementation on body weight, lipid and glucose metabolism in healthy individuals and those wit metabolic syndrome and type 2 diabetes.

It is a well-written manuscript and follows the guidelines for systematic reviews.

Reviewer 3 Report

The authors conducted systematic review and meta-analysis of the effect of resistant starch on cardio-metabolic health. The authors found that RS supplementation reduced triglycerides and weight though the significant effect on weight reduction might be due the results of a single study.

 Comments:

 I suggest to bring higher paragraph 5 of the introduction a higher. Resistant starch was defined in the fifth paragraph and it is not clear house the preceding paragraphs relate to RS.  

At the end of the introduction, you mentioned that the dose of RS administered to rodents far exceeded that could realistically be consumed by human. Do you mean in terms of relative to body surface area or in absolute amount? 

 In the methods you said: " During 123 the first-pass, article titles and abstracts were screened by two authors (JJ, DM, SK, AL, RS) ...' It appears that you listed five authors but said two authors. Which one is right? Check the statement that follows as well. 

 Metabolic syndrome is defined by 3 out of  5 criteria based on the values of blood pressure, blood glucose, triglyceride,  LDL, HDL and waist circumference. However, the authors list total cholesterol, HgA1c, insulin resistance, weight, satiety/appetite in addition to the above as metabolic outcome. What are you based on to define metabolic outcome?  

In the explanation of the mechanism of the RS benefits, have the authors considered the role of the fibers as food for the microbiota of the gut and their role in the gut homeostasis? Moreover, fibers have roles in the absorption of cholesterol. 

Have the authors investigated what might have contributed to the heterogeneity among the studies? I suggest to conduct meta-regression analysis of the given effect size as the dependent variable after a meta-analysis and study level characteristics as predictor variables for meta-analysis with significant I such as in Figure 5 and 8.

What do the double ended arrow and single ended arrows in table 1 indicate? Make sure to annotate tables and figures such that each is self-explanatory. A reader should not go else where to understand them.

Line 204: You may want to reconsider the statement, "No studies were able to detect any significant..."  for grammatical error.

 Line 245: What do you mean by "the publication bias was no longer prevalent"? 

It is not clear why the authors suggested the addition of short chain fatty-acids to the supplementation to achieve the hypothesized health benefits of RS. Do the authors mean that short chain fatty-acids are more produced in lab animals than human? Do the authors suggest to supplement the short chain fatty-acids to both experimental and placebo arms or to the experimental arm only?

Round 2

Reviewer 1 Report

The manuscript has substantially improved after revision. The conclusions are scientifically sound. However, I still feel too much data are presented (figures and tables) which is OK if the editorial policies allow this.

Few other points to consider

1-In the revised abstract (line 30-31): “All studies ranged from 1-12 weeks in duration, contained small sample sizes (10-60 participants) and most had an unclear risk of bias. Short-term”

while this statement is important but not suited for the abstract. You better move this to discussion or maybe into the limitation of the study.

2-in the introduction (line 83-85: The exact mechanisms ……. However, authors have suggested ……(Ref). 

 it would be better if changed as follow: The exact ..... However, it was suggested that......(Ref). 

3- Discussion (Line 389-391): The phrase “single studies” is not right “It should also be noted that …..by the results of single studies........”

Change into a few studies or a small number of studies.

4- Figures 2-9 could be minimized to highlight the important findings of the study (maybe a few of them only with proper figure’s legends to explain the data to the reader.

Reviewer 3 Report

The authors have addressed the comments in the first review.

For the description of article screeing, the authors may use:  Each article was screened by two of the listed authors (JJ, DM, SK, AL, RS). 
